# Targeting Endothelial HIF2α/ARNT Expression for Ischemic Heart Disease Therapy

**DOI:** 10.3390/biology12070995

**Published:** 2023-07-13

**Authors:** Karim Ullah, Lizhuo Ai, Zainab Humayun, Rongxue Wu

**Affiliations:** 1Section of Cardiology, Department of Medicine, Biological Sciences Division, University of Chicago, Chicago, IL 60637, USA; 2The Smidt Heart Institute, Cedars-Sinai Medical Center, Los Angeles, CA 90048, USA

**Keywords:** ischemic heart disease, endothelial cell function, HIF2α, ARNT, HIF pathway, inflammation

## Abstract

**Simple Summary:**

In this review paper, we discuss potential new therapies for ischemic heart disease (IHD), a leading cause of death and disability around the world. IHD results from a lack of oxygen supply to the heart muscle, often due to blocked or narrowed blood vessels. One key factor in IHD is the dysfunction of cells lining blood vessels, called endothelial cells. In response to low oxygen levels, these cells activate proteins called hypoxia-inducible factors (HIFs), which help regulate various processes related to cardiovascular diseases. In particular, we focus on HIF2α, which is mainly found in endothelial cells, and its partner ARNT, both of which play crucial roles in IHD development. Our review provides an overview of the current understanding of HIF2α and ARNT signaling in endothelial cells, their roles in inflammation and maintaining blood vessel integrity, and their involvement in IHD. We explore how these proteins work together and how their activity can be controlled using drugs. In conclusion, targeting HIF2α and ARNT in endothelial cells could offer promising new approaches for treating IHD, ultimately helping to improve the quality of life for millions of people affected by this debilitating condition.

**Abstract:**

Ischemic heart disease (IHD) is a major cause of mortality and morbidity worldwide, with novel therapeutic strategies urgently needed. Endothelial dysfunction is a hallmark of IHD, contributing to its development and progression. Hypoxia-inducible factors (HIFs) are transcription factors activated in response to low oxygen levels, playing crucial roles in various pathophysiological processes related to cardiovascular diseases. Among the HIF isoforms, HIF2α is predominantly expressed in cardiac vascular endothelial cells and has a key role in cardiovascular diseases. HIFβ, also known as ARNT, is the obligate binding partner of HIFα subunits and is necessary for HIFα’s transcriptional activity. ARNT itself plays an essential role in the development of the cardiovascular system, regulating angiogenesis, limiting inflammatory cytokine production, and protecting against cardiomyopathy. This review provides an overview of the current understanding of HIF2α and ARNT signaling in endothelial cell function and dysfunction and their involvement in IHD pathogenesis. We highlight their roles in inflammation and maintaining the integrity of the endothelial barrier, as well as their potential as therapeutic targets for IHD.

## 1. Introduction

Ischemic heart disease (IHD) is a prevalent condition resulting from insufficient oxygen supply to the heart muscle, causing significant morbidity and mortality worldwide [1,2,3]. Tissue survival and healthy organ function are dependent on adequate oxygen supply. Oxygen levels may also fluctuate in tissues, depending on the metabolic demand of the tissue and the oxygen tension in the blood. Therefore, local oxygen partial pressure acts as the main functional regulator of oxygen homeostasis. As the first responder to hypoxia, endothelial cells lining the microvasculature activate multiple signaling pathways to compensate for low oxygen tension, such as increased vasodilation and the expression of hypoxia-inducible factors (HIFs) [4,5]. HIFs are transcription factors that play a crucial role in cellular adaptation to hypoxia, and three well-recognized isoforms exist: HIF1α, HIF2α, and HIF3α, which are encoded by three distinct genes [6,7]. HIF1α and HIF2α share a similar domain composition [8], whereas HIF3α has high similarity in the bHLH and PAS domains with HIF1α and HIF2α but lacks the C-terminal transactivation domain (CTAD) [9]. HIF1α and HIF2α are the most well-studied members of this family and share a similar structure. Both are unstable under normal oxygen conditions (normoxia) due to the activity of prolyl hydroxylases (PHDs) that hydroxylate HIF1α and HIF2α, marking them for degradation. Under low oxygen conditions (hypoxia), the activity of PHDs is inhibited, leading to the stabilization of HIF1α and HIF2α. Upon stabilization, both HIF1α and HIF2α translocate to the nucleus, where they each dimerize with the aryl hydrocarbon receptor nuclear translocator (ARNT, also known as HIF1β). The resulting heterodimeric complexes, HIF1 (HIF1α/ARNT) and HIF2 (HIF2α/ARNT), then bind to hypoxia-response elements (HREs) in the promoters of target genes and activate their transcription. Although both HIF1 and HIF2 complexes bind to HREs and regulate gene expression in response to hypoxia, they are not identical in their functions. *Hif1α* gene transcriptions primarily govern metabolic reprogramming, while *Hif2α* plays a more significant role in regulating angiogenic extracellular signaling, guidance cues, and factors related to remodeling the extracellular matrix. Furthermore, HIF2α predominantly controls a wide range of transcription factors and coregulators, contributing to its diverse functions in hypoxic conditions [8,10]. HIF1α is the most studied, and HIF3α is the most novel, isoform. Only HIF2α is expressed in a more tissue-specific manner and highly expressed in endothelial cells [11,12,13]. Together, these findings support the critical role of HIF2α in vascular formation and function under hypoxic conditions. The HIF2α/ARNT complex plays critical roles in angiogenesis, anaerobic metabolism, and other processes in response to O_2_ deprivation [10]. This review provides an overview of the current understanding of the role of HIF2α (hypoxia-inducible factor 2α) and ARNT (aryl hydrocarbon receptor nuclear translocator) signaling in endothelial cell function and dysfunction, as well as their involvement in the pathogenesis of ischemic heart disease (IHD). The study aims to highlight the significance of endothelial dysfunction in IHD and the crucial roles played by HIF2α and ARNT in this process. In addition, this review aims to enhance the current knowledge of HIF2α and ARNT signaling in endothelial cells and their potential as therapeutic targets for ischemic heart disease. By examining these aspects, we hope to shed light on the pathogenesis of ischemic heart disease and explore the potential of targeting HIF2α and ARNT in endothelial cells as promising new approaches for treating IHD.

## 2. Activation of HIF during Hypoxia

Cardiac function relies on a consistent oxygen supply to meet metabolic demands. Oxygen deficiency, such as in coronary blockage or pressure overload states, leads the heart to initiate adaptive pathways to compensate. The hypoxia-inducible factor α-subunits (HIF-α) are key transcription factors involved in the mammalian response to low oxygen levels [14,15]. In oxygen-sufficient conditions, the expression of HIFα subunits is minimal due to rapid degradation; the catalytic domain of prolyl-4-hydroxylases (PHDs) recognizes a conserved LXXLAP (where X indicates any amino acid, and P indicates hydroxylacceptor proline) motif in the oxygen-dependent degradation domain (ODDD) of HIF-α subunits and hydroxylates specific proline residues of HIF-α subunits. The hydroxylated HIF-α subunits are recognized by a E3 ubiquitin ligase called the Von Hippel–Lindau protein (pVHL) and degraded by proteosomes [16].

HIFs are heterodimeric transcription factors comprising regulatory α (HIFα) and constitutive β (HIF1β) subunits (ARNT), which belong to the basic helix–loop–helix Per/Arnt/Sim (bHLH-PAS) superfamily of transcription factors [17,18]. Both HIFα and HIFβ (ARNT) proteins share a common structural bHLH domain responsible for dimerization and DNA binding [19]. Under hypoxic conditions, HIF1α and HIF2α escape from the proteasome-mediated degradation and bind to nuclear pore proteins via the nuclear localization signal (NLS) sequences. As a result, HIF-α subunits translocate into the nucleus, where they dimerize with ARNT (Figure 1), and the transcriptional coactivator CBP/p300 then binds to the CTAD of HIF-α subunits, which is required for the maximal activation of hypoxia-inducible genes [20,21]. HIF regulates a broad range of cellular processes, such as embryonic development, metabolic adaptation, angiogenesis, erythropoiesis, cell growth and differentiation, cell survival, and apoptosis [22,23]. In addition, the asparagine residues of HIF-α subunits are also hydroxylated by factors inhibiting HIFs (FIHs), which inhibits the binding of HIF with co-activators p300/CREB-binding protein [24,25]. During hypoxia, HIF2α/ARNT signaling activation is particularly vital in heart endothelial cells, as it promotes angiogenesis, regulates inflammation, maintains the endothelial barrier integrity, and prevents ischemic heart disease development and progression.

## 3. Role of HIF2α and ARNT in Vascular Endothelial Cells and Inflammation

### 3.1. HIF2a Expression and Inflammation

HIF2α, also known as endothelial PAS domain protein 1 (EPAS1), is abundantly expressed in vascular endothelial cells. HIF2α is a paralog of HIF1α (48% similarity) that also binds to ARNT [11]. Instead of being expressed in most cell types like HIF1α, the expression of HIF2α is confined to more specific cells and tissues, such as hepatocytes, cardiomyocytes, lungs, kidney interstitial cells, and some cell types in the central nervous system [11,26]. HIF2α binds to the hypoxia response element (HRE) and enhances the expression of genes for erythropoietin, vascular endothelial growth factor (VEGF), and various glycolytic enzymes, as well as activates the transcription of a reporter gene harboring the HRE [27]. Several studies have shown that HIF1α and HIF2α proteins are similarly induced by acute hypoxia in human lungs (4 h, 0.5% O_2_) at the translational or posttranslational level, but HIF1α protein stimulation disappears because of a reduction in its mRNA stability by prolonged hypoxia (12 h, 0.5% O_2_), whereas HIF2α protein stimulation remains high and stable during prolonged hypoxia [28]. Hif2α knockout mice experience perinatal deaths, but a small number of surviving Hif2α KO mice exhibit significant hematopoietic defects [29].

Inflammation and hypoxia are recognized as hallmarks of many pathological conditions and have been implicated in the pathogenesis of ischemic heart diseases. The presence of these conditions leads to the stabilization and activation of hypoxia-inducible factors (HIFs) in inflamed cells [30,31]. In a previously described ischemic kidney injury model, the authors found that endothelial Hif2α, but not Hif1α, regulated kidney inflammation via suppressing *Vcam1* expression [32]. Ischemia–reperfusion injury (IRI) of the kidneys revealed a significant increase of *Vcam1* mRNA expression in endothelial cell-specific *Hif2α* knockout kidneys, leading to prolonged leukocyte adhesion and transendothelial migration, which further exacerbates inflammation [32]. On the other hand, the *Hif2α*-dependent induction of amphiregulin expression results in faster recovery from myocardial IRI [33], and amphiregulin is known to suppress local inflammation [34]. Interestingly, global *Hif2α* knockdown-induced renal injury can be reversed by the restoration of HIF2α in ECs [32]. In lung endothelial cells, *Hif2α* deletion results in the increased expression of proinflammatory cytokines *Tnfα* and *Il1β*, whereas the overexpression of HIF2α limits SU5416-induced emphysema. SU5416 is a synthetic small molecule that acts as a selective inhibitor of vascular endothelial growth factor receptor-2 (VEGFR-2) tyrosine kinase activity [35].

Inflammatory stimuli such as LPS have been shown to significantly decrease HIF2α expression and increase the infiltration of inflammatory cells into the heart and the lungs. However, HIF2α induction through inactivating PHD2 reverses LPS-induced cardiac dysfunction [36]. Conversely, endothelial *Hif2α* deletion has been found to increase inflammation and induce lung vascular leakage [37]. It was reported that LPS stimulation enhances the expression of PHD2, leading to a decrease in the levels of Notch3 and HIF2α. However, this reduction in HIF2α and Notch3 expression was reversed by the overexpression of Sirtuin 3 [36]. Mechanistically, LPS stimulation suppresses the expression of Sirtuin-3, HIF2α, and Notch3 while promoting the expression of PHD2 and ang2 (Figure 2). Interestingly, the overexpression of Sirtuin 3 stimulates the expression of Ang-1/Tie-2 while reducing the expression of ang2 [38,39]. Additionally, Sirtuin 3 overexpression induces the expression of HIF2α and Notch3 [38,39]. Based on the above experimental evidence, it suggests that the induction of HIF2α may limit inflammation and promote endothelial cell survival by preserving the barrier integrity (Figure 2). Although the function of HIF2α in tissue inflammation has been studied in multiple organs and disease models, most studies suggest that HIF2α is protective in acute organ injuries but oncogenic during tumor development.

### 3.2. The Role of ARNT in Heterodimeric Transcription Factors and Endothelial Cell Function

The aryl hydrocarbon receptor nuclear translocator (ARNT), also known as HIF1β, is a transcription factor and belongs to the basic helix–loop–helix Per/Arnt/Sim (bHLH-PAS) superfamily of transcription factors [17]. Both the HIFα and HIF1β (ARNT) proteins share a common structural bHLH domain, which is responsible for dimerization and DNA binding [40]. As a NLS-containing transcription factor, ARNT was first identified as a factor required for the nuclear translocation of the ligand-bound aryl hydrocarbon receptor (AHR) [41]. ARNT is necessary for the generation of heterodimeric transcription factors with HIF-1 and HIF-2 alpha subunits and regulates the expression of genes involved in cell survival, proliferation, angiogenesis, and metabolism under hypoxia conditions [42,43]. For example, in mouse hepatoma cell lines, the induction of HIF target genes depends on the heterodimerization with ARNT [44], and *Arnt* knockout embryonic stem cells fail to induce the hypoxia-dependent expression of genes [45]. In primary endothelial cells, the loss of ARNT leads to reduced viability of the cells without affecting their proliferation [46].

### 3.3. The Anti-Inflammatory and Antioxidative Role of ARNT

Inflammation is a critical mediator in the pathogenesis of various cardiovascular diseases [47,48,49], and ARNT influences the expression of inflammatory cytokines [50]. The loss of ARNT is associated with an increase in the activity of NF-kB [51], a transcription factor that regulates the expression of genes involved in inflammation [52]. Mechanistically, ARNT interacts with RelB subunits of NF-kB and enhances the DNA-binding activity of RelB. The resulting ARNT/RelB complexes negatively regulate RelA/p50 complexes and suppress inflammation [52]. *Arnt* knockout in myeloid cells was found to increase inflammation, and these cells, when treated with LPS, showed an increase in the levels of proinflammatory cytokines, including IL-6 and TNF-a, pointing to the anti-inflammatory role of ARNT [53]. Further, ARNT inhibits inflammation in colitis, and ARNT repression by IFN-Y via JAK-3 signaling was associated with the flare-up of inflammation [54]. The protein AhRR, which represses AhR signaling by competing for binding to ARNT, suppresses the inflammatory response by forming a complex with ARNT [55,56]. Furthermore, ARNT decreases the production of reactive oxygen species (ROS) [57], which otherwise play a key role in mediating inflammation and endothelial barrier dysfunction [58,59,60], both of which lead to various cardiovascular disease pathologies, including atherosclerosis and ischemia–reperfusion injury [61,62,63]. A study in melanoma cells demonstrated that ARNT depletion results in increased ROS production due to alterations in mitochondrial mass and activity and decreased NQO1 expression, an antioxidant [64]. Another study showed that overexpression of ARNT increases the levels of antioxidant glutathione in AML cells, thereby conferring protection against the damaging effects of ROS by scavenging them [65]. ARNT is also linked to the risk of tumor progression, tumor invasion, and metastasis. A decrease in ARNT expression helps in the spread of cancer cells, particularly in melanoma [64]. The inactivation of endothelial ARNT increases reactive oxygen species (ROS) in high glucose conditions, which impairs cellular function in several organs, including the heart, retina, kidneys, and nervous system [43,66]. When the Arnt gene is suppressed, there is a significant increase in the levels of ROS [43,57]. The above experimental evidence suggests that ARNT reduces ROS production through multiple mechanisms, including the regulation of mitochondrial function, NQO1 expression, and modulation of antioxidant levels like glutathione. Furthermore, the suppression of NF-kB activation via ARNT [51] helps to mitigate inflammation, which, in turn, reduces ROS generation. These combined effects contribute to thhe protective role of ARNT in preventing inflammation-related cardiovascular diseases. These findings collectively suggest that ARNT plays a protective role in mitigating inflammation by inhibiting the NF-kB pathway and regulating ROS production, potentially preventing the development of inflammation-related cardiovascular diseases.

## 4. Potential Role of HIF2α and ARNT in Ischemic Heart Disease

### 4.1. HIF2α and ARNT in Mouse Heart Development and Cardiovascular Function

Mouse heart development occurs between embryonic days E7.75–15, with various cells of different origins contributing to heart development [67]. During embryogenesis, the rapid growth of embryonic tissue increases oxygen consumption, creating a hypoxic microenvironment [68] and activating the components of HIF (HIF1α, HIF2α, and ARNT) [69,70]. Vascular endothelial cells are fundamentally important for heart development and are associated with both surfaces of the myocardium. The myocardium is generally hypoxic during the early stage of development [67,71]. Vascular endothelial cell and myocardium development depend on a reciprocal interaction [72,73]. The loss of HIF activity results in both myocardial and endocardial defects during development. In vivo studies have suggested that Hif1α is essential in the myocardium but not in the endothelium for normal development of the heart [67]. Mice lacking functional Hif2α in vascular endothelial cells develop multiple cardiovascular defects, including a thin myocardium, a disorganized endocardium, and irregular trabeculation [74].

Mouse genetic studies have revealed the critical role of HIF signaling pathways in vascular development and pathogenesis. The genetic inactivation of *Hif1α* or *Arnt* results in embryonic lethality due to abnormal vascular development, while the inhibition of *Hif2α* leads to impairment in vascular remodeling and an altered cardiac rhythm, depending on the genetic background of the mice [75,76]. HIF isoforms are expressed at different heart developmental stages in different cell types [77]. HIF1α is mainly expressed in the cardiomyocyte, while HIF2α is predominantly expressed in endothelial cells in the heart during embryonic development [78]. *Hif1α+/−* mice develop normally but show impaired physiological responses to chronic hypoxia. One strain of Hif2α-deficient mice displayed hemorrhage and failed to maintain discrete vascular tubular structures, indicating Hif2α’s essential role in vascular remodeling during development.

In addition to embryonic development, Hif2α plays a key role in ischemic heart disease (IHD). IHD is a condition where the blood supply to the heart muscle is reduced, typically due to atherosclerosis or coronary artery blockage, leading to inadequate oxygen supply to the heart tissue [79]. HIF2α expression in vascular endothelial cells during IHD has several functional consequences. For example, myocyte-specific *Hif2α* deletion enhances ischemic injury in mouse models [33]. One study showed that EC-specific Hif2α deletion in mice impaired angiogenesis in its hindlimb ischemia and autochthonous solid tumor model [80,81]. Under hypoxic conditions, it has been observed that the HIF2α isoform plays a pivotal role in regulating the expression of the *DiI4*, *Adm1*, and *Ang2* genes [80]. This indicates that *Hif2α* is critically involved in facilitating complementary angiogenesis within ischemic tissue at a mechanistic level. In addition, HIF2α induces the expression of proangiogenic factors such as vascular endothelial growth factor (VEGF) [82]. The induction of VEGF stimulates the growth of new blood vessels and facilitates blood flow to ischemic tissues [83]. A recent study revealed that the induction of HIF2α promotes endothelial cell survival under hypoxia and maintains the endothelial barrier integrity [37,81,84]. These findings suggest that HIF2A enhances endothelial cell survival under ischemic conditions. In summary, HIF2α is not only critical for embryonic development but also plays a key role in the pathogenesis of ischemic heart disease. Its involvement in promoting angiogenesis, inducing the expression of proangiogenic factors, and enhancing endothelial cell survival under hypoxia highlights its potential as a therapeutic target for the treatment of IHD.

Impaired angiogenesis and vascular barrier dysfunction contribute to cardiovascular diseases [85,86,87,88,89]. The endothelial-specific expression of ARNT in mice is crucial for the formation of blood vessels [90], as well as the normal development and functioning of the heart [91,92]. A recent study on zebrafish revealed that mutations in both ARNT 1 and ARNT 2 in the embryo inhibited the formation of blood cells and interfered with endothelial cell formation as well [93]. The loss of endothelial ARNT also resulted in severe bleeding in the heart, and nearly 90% of the mice embryos could not survive beyond 10.5 days [92]. Another study reported similar results and attributed the lethality to defective placental vascularization [94]. *Arnt* deletion affects vascularization in the yolk sac, and the defects in new blood vessel formation resemble those in mice lacking VEGF [45]. The loss of *Arnt* affects the development of vasculature by altering and disrupting tube formation and decreasing the expression of the genes involved in the formation of new vessels, such as VEGF-A [95]. Therefore, endothelial cell-specific ARNT expression is crucial for angiogenesis and may protect against ischemic heart diseases.

### 4.2. Cardioprotective Potential of ARNT and HIF2α

ARNT plays a significant role in preventing the development of cardiomyopathy and subsequent heart failure. The cardiac-specific deletion of *Arnt* in mice leads to PPARα activation, dilated left ventricular chambers, and impaired cardiac contractility, resembling diabetic cardiomyopathy [96]. Furthermore, the endothelial cell-specific deletion of *Arnt* results in increased cardiomyocyte size and impaired cardiac contractility [97]. ARNT also protects against cardiac, renal, and liver fibrosis by acting through the FKBP-YY1-ARNT-ALK3 signaling pathway [98]. The deletion of *Arnt* promotes fibrosis by increasing collagen 1A1 and MMP9 production [53]. The inhibition of FKBP12/YY1 increased ARNT expression, reducing organ damage in renal, heart, and liver fibrosis models [99,100]. These findings underscore ARNT’s protective role in the development and progression of heart failure, making it a potential therapeutic target for ischemic heart failure and the treatment for maladaptive cardiac fibrosis in chronic inflammatory heart diseases.

HIF2α is highly expressed in vascular endothelial cells and regulates the expression of target genes responsible for vascular function and angiogenesis [80]. Population genetic and animal model studies suggest that HIF2α is a critical regulator of several cardiovascular diseases [77,95,96]. Indeed, HIF2α plays a crucial role in maintaining vascular integrity. For example, endothelial cell-specific HIF2α deletion increases vascular permeability in multiple organs, including the lungs [80]. In ischemia–reperfusion injury of the kidneys, activation of HIF2α by using PHD inhibitor FG4487 protects against renal failure [101]. Similarly, the use of l-mimosine and dimethyloxalylglycine, which inactivate prolyl-4 hydroxylases, thereby activating HIF signaling, protects mouse kidneys from injury induced by ischemia–reperfusion in the kidneys [102]. HIF is considered a central component of ischemic preconditioning of the heart, in which repetitive short exposure to ischemia and reperfusion is given before a subsequent long exposure to ischemia and helps to preadapt the myocardium to activate protective mechanisms. Several studies have shown the cardioprotective effects of HIF activation during ischemic injury [103]. The induction of HIF2α provides protective mechanisms in cardiomyocytes’ adaptation to chronic hypoxia [104]. Furthermore, the deletion of HIF2α, rather than HIF1α, demonstrates the essential role of HIF2α in myocytes for improving the tolerance to myocardial ischemia–reperfusion injury. This is achieved through the upregulation of its specific target gene, amphiregulin [33]. In contrast, the induction of HIF2α by inactivating PHD2 protects the mouse heart from acute ischemia–reperfusion injury [105]. As HIF2α is highly expressed in endothelial cells, the activation of HIF2α in these cells, either by inactivating HIF2α-degrading enzymes, such as PHD2, or by overexpressing HIF2α, could provide a novel therapeutic approach for the treatment of cardiovascular disease. However, the role of endothelial HIF2α in ischemic heart diseases requires further investigation.

## 5. Future Perspectives and Limitations

The emerging evidence discussed in this review highlights the potential of targeting endothelial HIF2α/ARNT expression as a therapeutic approach for ischemic heart disease. HIF2α has been found to be endothelial cell-specific and has distinct protective signaling pathways independent of other HIF isoforms. It has shown promising therapeutic value for the treatment of different cardiovascular diseases. Extensive studies have been carried out using different mouse models to precisely characterize the role of HIF2α in different tissues. However, many questions remain unanswered regarding the underlying mechanisms of the HIF2α-mediated regulation of signaling pathways in cardiovascular diseases. Elucidating the precise molecular mechanisms and signaling pathways underlying the cardioprotective effects of endothelial HIF2α/ARNT is crucial, as this knowledge will help in the development of more targeted and effective therapies. The crystallographic structures of the HIF2α/ARNT complex have provided insights into the dimer architecture and the mechanism of inhibition by small molecules [106], allowing for the development of potential therapeutics targeting HIF2α/ARNT signaling and for future research and drug development. Investigating potential side effects and safety concerns associated with modulating endothelial HIF2α/ARNT expression is necessary to ensure clinical feasibility. Exploring the possibility of combinatorial therapies involving targeting endothelial HIF2α/ARNT, along with other known therapeutic targets in ischemic heart disease, may lead to more effective treatment strategies. Further research should focus on developing novel pharmacological agents or gene therapy approaches that can specifically target endothelial HIF2α/ARNT expression in the heart while minimizing off-target effects. Finally, future clinical trials are needed to evaluate the efficacy and safety of therapies targeting endothelial HIF2α/ARNT in patients with ischemic heart disease and identify the optimal patient population that may benefit from such treatments.

While targeting HIF2α and ARNT holds considerable potential in the treatment of ischemic heart disease (IHD), several challenges and limitations must be acknowledged. First, the specificity in targeting HIF2α and ARNT remains a concern. Given their broad involvement in various physiological processes beyond the cardiovascular system, the systemic modulation of these factors could inadvertently affect non-target tissues or systems, leading to potential off-target effects or side effects. Targeting endothelial HIF2A-ARNT poses additional challenges in terms of delivering therapeutic agents to these specific cells within the target organ. Endothelial cells are not the same in all organs. They differ in terms of their structure, function, and surface markers based on their location in the body. This makes it challenging to design a universal drug delivery system that can target endothelial cells across different organs effectively. One major limitation in targeting organ endothelial cells is the lack of affinity between the therapeutic agents and the endothelial cells. Another limitation is the difficulty in the developing methods that specifically target and deliver therapeutic agents to endothelial cells of a target organ. Overcoming these limitations is an active area of research, and scientists are exploring various strategies to improve targeted delivery methods to organ endothelial cells. Second, the temporal aspect of HIF2α and ARNT activation in relation to disease progression is another factor that complicates the therapeutic design. It is crucial to determine the ideal window for therapeutic interventions, as the consequences of modulating HIF2α and ARNT might vary depending on the disease stage. Third, the complexity and redundancy of hypoxia signaling pathways can pose challenges. HIF2α and ARNT are part of a complex network of hypoxia-responsive elements, and the disruption of one element may be compensated by others. This resilience can potentially reduce the effectiveness of therapies targeting HIF2α and ARNT alone. Lastly, individual patient variations in hypoxia response and the genetic and epigenetic factors influencing HIF2α and ARNT expression can impact the efficacy of therapeutic strategies. More extensive research is needed to address these challenges and to refine the therapeutic strategies targeting HIF2α and ARNT in the context of IHD. In conclusion, targeting endothelial HIF2α/ARNT expression holds promise as a potential therapeutic strategy for ischemic heart disease, and further research in this area will be instrumental in advancing our understanding of HIF2α/ARNT’s role in ischemic heart disease pathophysiology and developing novel, effective treatment options for patients.

## Figures and Tables

**Figure 1 biology-12-00995-f001:**
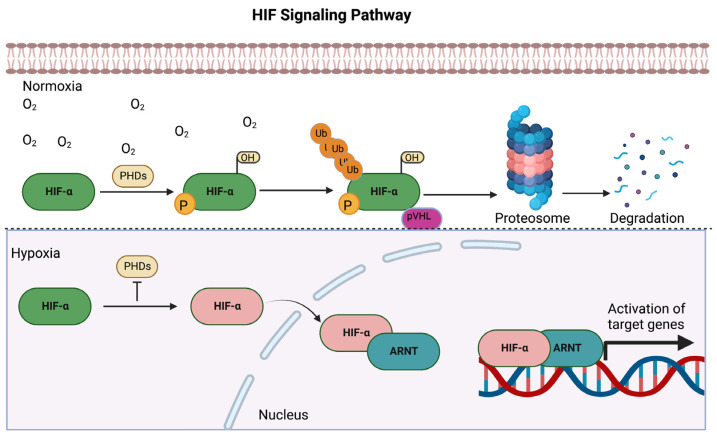
Schematic representation of the hypoxia response pathway. In normal conditions, prolyl hydroxylases (PHDs) catalyze the hydroxylation of HIFα subunits, tag them for ubiquitination, and subsequently degrade them by the proteosome. While under hypoxic conditions, the activity of PHDs is inhibited, and HIF-α subunits escape from ubiquitin-mediated proteasomal degradation and translocate into the nucleus, where it initiates the transcription of target genes to maintain oxygen homeostasis.

**Figure 2 biology-12-00995-f002:**
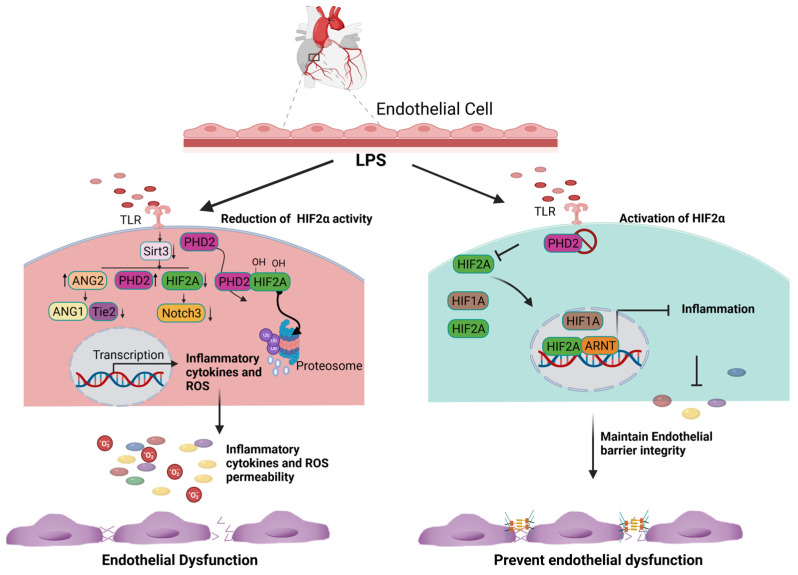
Cardioprotective role of endothelial HIF2A expression. LPS stimulation increases the expression of PHD2 and limits the amount of HIF2A via the hydroxylation-dependent degradation of HIF2A. In the absence of HIF2A, inflammatory genes are transcribed and generate excessive ROS. Consequently, it increases endothelial cell apoptosis and loss of the endothelial barrier function (Right panel). The inhibition of PHD2 results in the accumulation and translocation of HIF2A into the nucleus. HIF2A binds to ARNT, limits the expression of inflammatory genes, and subsequently decreases ROS generation. The inhibition of ROS production and inflammation prevent endothelial barrier dysfunction.

## Data Availability

No new data were created.

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
