# Peer review of "Targeting Endothelial HIF2α/ARNT Expression for Ischemic Heart Disease Therapy"

_biology, 2023, doi:10.3390/biology12070995_

Round 1

Reviewer 1 Report

In this review, the authors highlight the current understanding of HIF2α and ARNT signaling in endothelial cells and their role in inflammation and maintenance of vascular integrity, as well as their involvement in IHD. Based on all the emerging evidence discussed in this review, the potential of targeting endothelial cell HIF2α/ARNT expression as a therapeutic approach in ischemic heart disease is highlighted. Overall, this is an interesting summary and review, but there are still several concerns need to be clarified by the authors.

1.     The first illustration before the introduction is confused: a) Is the role of HIF1α in ischemic heart disease the opposite of HIF2α? However, from the published references, it appears that HIF1α and HIF2α play the same protective role in ischemic heart disease, and the authors need to clarify. b) this illustration is redundant compared to Figure 2.

2.     Authors are advised to go through the manuscript again carefully to check for typos, grammatical errors, and redundant sentence. For example, line128 to line131.

3.     In line254, “In addition, it has been shown that deletion of HIF2α and not HIF1α in myocytes enhances tolerance to myocardial ischemia-reperfusion injury by increasing the expression of its target gene amphiregulin [32]”. This statement means the exact opposite of what is stated in the reference.

4.     Please give the full name/description of the abbreviation when it first been descripted such as “CTAD”, et al.

5.     In line 111, “Hif2α but not HIF1α” the format of the gene name needs to be consistent.

6.     Where is the figure 3, line136.

Authors are advised to go through the manuscript again carefully to check for typos, grammatical errors, and redundant sentence. For example, line128 to line131.

Author Response

Reviewer's comment: In this review, the authors highlight the current understanding of HIF2α and ARNT signaling in endothelial cells and their role in inflammation and maintenance of vascular integrity, as well as their involvement in IHD. Based on all the emerging evidence discussed in this review, the potential of targeting endothelial cell HIF2α/ARNT expression as a therapeutic approach in ischemic heart disease is highlighted. Overall, this is an interesting summary and review, but there are still several concerns that need to be clarified by the authors.
The first illustration before the introduction is confusing: a) Is the role of HIF1α in ischemic heart disease the opposite of HIF2α? However, from the published references, it appears that HIF1α and HIF2α play the same protective role in ischemic heart disease, and the authors need to clarify. b) this illustration is redundant compared to Figure 2.
Response: Thank you for your astute observations and for pointing out the confusion in our manuscript. The illustration before the introduction was intended as a graphical abstract, but it appears its placement prior to the introduction has caused confusion. To avoid any further redundancy and confusion, we have decided to remove the initial illustration and retain Figure 2, which provides a more detailed representation of our focus.
Regarding a) Is the role of HIF1α in ischemic heart disease the opposite of HIF2α? The roles of HIF1α and HIF2α in ischemic heart disease are indeed complex and can vary depending on the context, and we have clarified this in the revised manuscript. Response: HIF-1α and HIF-2α show different specificity in their transcriptional targets. For instance, HIF-1α effectively stimulates the expression of glycolytic enzymes, such as Lactate dehydrogenase-A (LDH-A) and CA IX. In contrast, HIF-2α acts more effectively on the EPO gene (encodes for erythropoietin) and genes involved in iron metabolism, while another group of genes, including VEGF and GLUT-1, are regulated by both HIF-1α and HIF-2α [1] we have included the citation in the manuscript.
Authors are advised to go through the manuscript again carefully to check for typos, grammatical errors, and redundant sentences. For example, line 128 to line 131.
Response: Thank you, and the manuscript has been thoroughly reviewed.
In line 254,

In addition, it has been shown that deletion of HIF2α and not HIF1α in myocytes enhances tolerance to myocardial ischemia-reperfusion injury by increasing the expression of its target gene amphiregulin [32]". This statement means the exact opposite of what is stated in the reference.
Response: Thank you for bringing this to our attention. We apologize for the error. The statement has been corrected to reflect the findings in the reference accurately.
Please give the full name/description of the abbreviation when it first been described, such as "CTAD," et al.
Response: We appreciate your suggestion, and we have made the necessary revisions to provide the full name/description of abbreviations upon their first mention in the revised manuscript.
In line 111, "Hif2α but not HIF1α," the format of the gene name needs to be consistent.
Response: Thank you for catching that inconsistency. The format of the gene name has been corrected for consistency in the revised manuscript.
Where is figure 3, line 136?
Response: We apologize for the oversight. The missing reference to Figure 3 is corrected

Reviewer 2 Report

Overall, this review article effectively summarizes the importance of endothelial dysfunction, hypoxia-inducible factors (HIFs), and their role in ischemic heart disease (IHD). The review article focuses on the roles of HIF2α and ARNT in the endothelial response to hypoxia and inflammation. While the topic is relevant and important, it would be helpful to clearly state the specific objectives of the review article to guide the readers throughout the manuscript.

1) The introduction states that HIF2α is considered the primary regulator of the endothelial response to hypoxia, but this claim requires further support. Please provide relevant references or evidence to support this statement.

2) In the section of line 140-141, please provide more specific information about the functional consequences of HIF2α expression in vascular endothelial cells during IHD. This would clarify its diverse roles in different contexts.

3) In section C of line 155, please provide more evidence and discuss the specific mechanisms by which ARNT decreases the production of ROS, inhibits inflammation and its interaction with NF-κB in greater detail

It would be helpful to include a brief mention of the potential limitations or challenges in targeting HIF2α and ARNT for therapeutic interventions in IHD.

Minor comments:

1.       The graphical abstract figure requires clarification regarding the interactions between HIF2A and ARNT in the context of ischemic/hypoxia-related vascular dysfunction. Please amend the figure to provide a clearer representation.

2.       The symbol for the superoxide ion in the graphical abstract appears to be incorrect. Please correct this error.

3.       Please ensure that all abbreviations used in the article are defined when only first mentioned. For example, LXXLAP should be defined upon its first occurrence (line 69).

4.       Please ensure consistent formatting throughout the manuscript, such as italicize "genes" in all contexts.

The quality of English language in the manuscript is generally good. The writing is clear and coherent, with a few minor grammatical errors and inconsistencies.

Author Response

Comment: Overall, this review article effectively summarizes the importance of endothelial dysfunction, hypoxia-inducible factors (HIFs), and their role in ischemic heart disease (IHD). The review article focuses on the roles of HIF2α and ARNT in the endothelial response to hypoxia and inflammation. While the topic is relevant and important, it would be helpful to clearly state the specific objectives of the review article to guide the readers throughout the manuscript.

Response: We appreciate your recognition of the importance of our topic and your suggestion for improvement. We have revised the introduction to more explicitly state the specific objectives of the review, offering a clear roadmap for readers to navigate our manuscript.

Comment: To explore potential therapeutic interventions targeting HIF2α and ARNT for the treatment of ischemic heart disease. The introduction states that HIF2α is considered the primary regulator of the endothelial response to hypoxia, but this claim requires further support. Please provide relevant references or evidence to support this statement.

Response: Thank you for pointing out the need for additional references to substantiate our claim regarding HIF2α. We have now included additional sources highlighting the crucial role of HIF2α in modulating the endothelial response to hypoxia.

Minor Comments:

Comment:

  1. The graphical abstract figure requires clarification regarding the interactions between HIF2A and ARNT in the context of ischemic/hypoxia-related vascular dysfunction. Please amend the figure to provide a clearer representation.
  2. The symbol for the superoxide ion in the graphical abstract appears to be incorrect. Please correct this error.
  3. Please ensure that all abbreviations used in the article are defined when only first mentioned. For example, LXXLAP should be defined upon its first occurrence (line 69).
  4. Please ensure consistent formatting throughout the manuscript, such as italicize "genes" in all contexts.

Response: We are grateful for your detailed review and attention to these elements. Regarding the graphical abstract, given the similarity between it and Figure 2, we've decided to remove the graphical abstract to avoid confusion, and we've ensured that Figure 2 provides a clear representation of HIF2A and ARNT interactions. We have also corrected the symbol for the superoxide ion. We have also revised the manuscript to define all abbreviations upon their first use and apply consistent formatting, including italicizing gene names. 

Round 2

Reviewer 1 Report

The manuscript is a marked improvement over the previous version in terms of writing and arrangement of data. The authors have responded appropriately to most of the issues raised by the reviewer. There are some minor adjustments that need to be made by the authors:

“The limitation may also include specific and effective methods of organ endothelial delivery”.

Author Response

Thank you very much for pointing out that specific and effective methods of organ endothelial delivery represent another potential limitation of our study. The strategies for targeted drug delivery to endothelial cells are complex and continually evolving. We agree that our manuscript could benefit from a discussion about this aspect, and we have now included these limitations accordingly.